computer modelling and simulation/statistics

uncertainty, biophysical modelling, interdisciplinary research, social science, statistics, engineering

**Author for correspondence:**
M. Espig
e-mail: martin.espig@agresearch.co.nz

# Uncertainty in and around biophysical modelling: insights from interdisciplinary research on agricultural digitalization

M. Espig[1], S. C. Finlay-Smits[1], E. D. Meenken[1], D. M. Wheeler[2] and M. Sharifi[1]

[1]AgResearch, Lincoln Research Centre, 1365 Springs Road, Lincoln 7674, New Zealand
[2]AgResearch, Ruakura Agricultural Centre, 10 Bisley Road, Enderley, Hamilton 3214, New Zealand

ME, 0000-0002-0682-1999; SCF-S, 0000-0002-6331-9284; EDM, 0000-0001-6715-1653; MS, 0000-0002-3340-4808

Agricultural digitalization is providing growing amounts of real-time digital data. Biophysical simulation models can help interpret these data. However, these models are subject to complex uncertainties, which has prompted calls for interdisciplinary research to better understand and communicate modelling uncertainties and their impact on decision-making. This article develops two corresponding insights from an interdisciplinary project in a New Zealand agricultural research organization. First, we expand on a recent *Royal Society Open Science* journal article (van der Bles *et al.* 2019 *Royal Society Open Science* **6**, 181870 (doi:10.1098/rsos.181870)) and suggest a threefold conceptual framework to describe direct, indirect and contextual uncertainties associated with biophysical models. Second, we reflect on the process of developing this framework to highlight challenges to successful collaboration and the importance of a deeper engagement with interdisciplinarity. This includes resolving often unequal disciplinary standings and the need for early collaborative problem framing. We propose that both insights are complementary and informative to researchers and practitioners in the field of modelling uncertainty as well as to those interested in interdisciplinary environmental research generally. The article concludes by outlining limitations of interdisciplinary research and a shift towards transdisciplinarity that also includes non-scientists. Such a shift is crucial to holistically address uncertainties associated with biophysical modelling and to realize the full potential of agricultural digitalization.

# 1. Introduction

The environmental regulation of industrialized agricultural systems is increasingly driven by information derived from digital data that are generated by smart sensors and Internet-of-Things devices, as well as advanced analytics processing these data. Digitalization also changes on- and off-farm decision-making by providing high volumes of almost real-time data to farmers in smart farming operations, consumers and others along the supply chain [1–3]. Computer simulation models help to interpret the growing amount of data, support agricultural decision-making and inform environmental regulation [4–7]. While models may help to reduce some uncertainties, they cannot address others and might introduce new uncertainties. Both modellers and model users frequently refer to uncertainty, but it is often unclear which specific aspects of a model are deemed 'uncertain' or whether discussions even relate to the actual modelling. Communication around the uncertainties associated with modelling (hereafter 'modelling uncertainty') can consequently remain vague and ineffective, which poses challenges for realizing the full potential and societal benefits of agricultural digitalization [1,8].

A growing body of research is investigating the intricacies of modelling uncertainty [9,10], as well as their effective framing and communication [11,12]. Given the pivotal role of computer simulation models in environmental and agricultural decision-making, this research has started to generate much needed clarity around uncertainties associated with these models. Yet, modelling uncertainties can emerge from diverse sources such as sensor measurement errors, the inherent unpredictability of some biophysical processes, statistical analysis and human (mis)understanding. Uncertainties might also be associated with the generation and application of modelled information within particular decision-making or policy contexts. Discerning different quantitative and qualitative modelling uncertainty can, therefore, be difficult (e.g. [13]). While the interdisciplinary nature of uncertainty has been recognized [14], developing holistic perspectives that account for the various sociocultural and biophysical facets of modelling uncertainties remains challenging and requires meaningful collaboration between researchers in the social, natural and applied sciences.

This article reflects on an interdisciplinary research programme that investigates uncertainties in sensor measurements and biophysical models (a subset of computer simulation models) associated with the digital transformation of New Zealand's agricultural sector. Based at AgResearch, one of New Zealand's Crown Research Institutes, the research team includes members with expertise in anthropology, biophysical sciences, data science, engineering, modelling and statistics. A lack of shared uncertainty definitions initially constituted a major challenge for the team. Following our experiences of addressing this challenge, we present insights that should be informative to researchers and practitioners in the field of modelling uncertainty, regulators and policymakers, as well as to those interested in interdisciplinary environmental research more generally. As such, the article is centred around an overview of various qualitative and quantitative uncertainties in and around computer simulation models as the object of study. We present a conceptual framework to capture the multitude of disciplinary insights into these uncertainties. While by no means exhaustive, our framework contributes to the growing body of modelling uncertainty research. In reflecting on the development of this framework, we then shift from the object to the process of research by using our programme to illustrate the framework's usefulness in establishing interdisciplinary approaches. These reflections highlight that forming interdisciplinary collaboration and building mutual respect for different ontologies, epistemologies and methodologies are indispensable steps in systematically addressing modelling uncertainties. However, putting these considerations into practice involves organizational and behavioural challenges. Our interwoven accounts on both the object and process of research on modelling uncertainty in the context of agricultural digitalization should thus be seen as complementary. The article's main contribution is, therefore, to demonstrate that holistically understanding modelling uncertainties requires meaningful collaboration between researchers from a range of disciplines. However, establishing successful interdisciplinary approaches is usually a laborious process that involves overcoming numerous practical and conceptual challenges. We present these insights to prompt discussion among researchers and practitioners about how modelling uncertainties may be addressed in a systematic fashion and to reflect on the underlying processes of interdisciplinary research, which are often overlooked.

The next section reflects on interdisciplinary agricultural research and the challenges of equal disciplinary standing to contextualize our insights. The third section outlines three distinct disciplinary perspectives on uncertainties associated with modelling and sensor measurements in digital agriculture innovation. These three perspectives highlight the diverse aspects that warrant consideration when

seeking to address modelling uncertainties and demonstrate differences in underlying ontologies and epistemologies. The fourth section introduces a conceptual framework our team co-developed to bridge these differences and to conceive the notion of modelling uncertainties in an inclusive way. This framework is centred on a distinction between direct, indirect and contextual uncertainties. We then reflect on the framework's usefulness for shaping our interdisciplinary project and argue that it allowed us to collaboratively frame problems associated with understanding modelling uncertainties, thereby ordering and delineating our individual research contributions. However, this process included several challenges that require critical reflection and a deeper engagement with interdisciplinarity as a way of conducting collaborative research. We conclude by summarizing the complementary insights into both the object and process of research and consider the potential limitations of interdisciplinary approaches. This reflection points towards transdisciplinary research and collaborative problem framing that also includes non-scientific actors and diverse forms of knowledge.[1]

# 2. Interdisciplinary agricultural research

A central aim of our research has been the formation of an interdisciplinary team to understand uncertainties associated with the digitalization of New Zealand's agricultural sector. On the one hand, this was prompted by organizational requirements that follow the now commonplace notion of problem-oriented environmental research requiring innovative approaches beyond disciplinary boundaries [1,16]. Complementarity and disciplinary integration are central components of AgResearch's *New Zealand's Bioeconomy in the Digital Age* (NZBIDA) platform, within which our research has been conducted and which is one of the organization's flagship programmes for developing a new *modus operandi* to bridge disciplinary silos. On the other hand, interdisciplinarity is a well-established response to an evolving relationship between science and wider society, with historically established barriers between them breaking down due to increased societal demands for transparent, open and accountable research [17,18]. Like many research organizations, we, therefore, strive for responsible research and innovation [19,20] and explore novel ways of conducting scientific research. Barry & Born [21] describe both of these aspects as following logics of *innovation* and *accountability*. Within these logics, interdisciplinarity might be regarded as a response to societal challenges and an improved value proposition to funding agencies. It is thus important to consider the contexts in which interdisciplinary research of modelling uncertainties takes place in order to assess, and improve, projects' efficacy.

Establishing successful interdisciplinary collaboration beyond engagements of multiple disciplines can be challenging in agricultural research organizations that often continue to be dominated by biophysical scientists, partly also due to funding models that historically lacked incentives for collaboration across disciplines. What constitutes success may differ between interdisciplinary team members or supervising research managers, particularly if differences in underlying ontologies, methodologies and criteria for the validity of generated knowledge are not explicitly negotiated. Equity in researchers' roles and the status of their disciplinary backgrounds can, therefore, not be taken for granted. Verma *et al.* [22], for instance, describe how sociocultural research at the Consultative Group for International Agricultural Research has historically been marginalized and regarded as an auxiliary support service to the biophysical sciences, driven by discourses of technically and economically oriented 'solutions' to agricultural issues. Such unequal disciplinary relations are often rooted in a perceived nature–culture dichotomy that suggests fundamental divisions between the 'hard' natural sciences and 'soft' social research (cf. [23]). However, these categorical distinctions are misleading, as we argue below with respect to the interconnected uncertainties associated with biophysical modelling and socio-technical–environmental systems generally. Disciplinary divisions can also not only be observed between the social and natural sciences, as those of us working in statistics and engineering have, in some instances, experienced similar perceptions of their discipline as fulfilling mere support functions.

Sociocultural researchers' calls to address this imbalance [24] and to 'keep culture in agriculture' [25] are not intended as naive critiques of the important contribution of the biophysical sciences in

---

[1]Multi-, inter- or transdisciplinary approaches are increasingly used to facilitate integrated research beyond single disciplines. We follow Stock & Burton [15, p. 1096] in that '[i]nterdisciplinary studies focus on addressing specific "real world" system problems and, as a result, the research process forces participants (from a variety of unrelated disciplines) to cross boundaries to create new knowledge'. Transdisciplinary research takes integration and cooperation further and involves 'not only multiple disciplines, but also multiple *non-academic participants* (e.g. land managers, user groups, the general public) in a manner that combines interdisciplinarity with participatory approaches' ([15, p. 1098], original emphasis).

agricultural and natural resource management. However, acknowledging the often persistent dominance of the biophysical sciences in agricultural research is a crucial prerequisite for successful interdisciplinary research. As such, notions of inter-, multi- or transdisciplinarity point towards an institutional fragmentation of knowledge production within many research organizations [15,16]. This fragmentation is reproduced in everyday research where disciplines are 'ways of keeping distinct the origins not just of ideas and materials but of work practices, the lines of authentication and accountability' [18, p. 131]. Disciplinary divisions may also maintain the status of some branches of science and researchers. Considering not only the institutional arrangements but also everyday research practices that re-create disciplinary distinctions, including their potentially unequal relationships, highlights that meaningful interdisciplinarity cannot be realized through organizational policies or institutional research strategies alone. It must also become embedded in researchers' routine practices and interactions. This requires cultivating an appreciation for different disciplinary insights that are of equal epistemic standing, rather than regarding some disciplines as merely 'supporting' core research. Otherwise, notions of interdisciplinarity might remain well-meaning aspirations or, worse still, window-dressing rhetoric.

Verma *et al*. [22] argue that, due to their often marginal status, sociocultural researchers in biophysically dominated agricultural research environments become accustomed to learning from colleagues in the natural sciences and to adopting their disciplinary languages in (re-)framing their knowledge claims. However, interdisciplinary research is ultimately unlikely to be successful if the intellectual responsibilities for being collaborative primarily lie with any one group or individual, regardless of disciplinary background. Instead, diverse contributions must be equally valued by all parties, which include reflecting on disciplinary strengths and limitations [26]. Producing mutually comprehensible knowledge is thus the responsibility of all researchers in collaborative settings. This includes not only regarding interdisciplinary research within logics of innovation and accountability but also aiming for a deeper 'cultivation of interdisciplinary subjectivities and skills' [21, p. 39]. Such an interdisciplinary logic of *ontology* directs one 'to consider the diverse ways in which the reconfiguration of the relations between the social and natural sciences is today being posed anew' [21, p. 42]. We regard this type of deeper engagement with interdisciplinary research as an important component of programmes that seek to address the various facets of modelling uncertainties.

Interdisciplinarity can thus transcend merely functional logics and form part of a more profound reorientation towards integrated research with its potential to foster, and embrace, explanatory pluralism [27,28]. In this sense, interdisciplinary research should not be an ad hoc approach chosen once a problem has been delineated. Instead, it can form the very foundation upon which research projects are built. Brondizio [16], therefore, regards interdisciplinarity as collaborative problem framing and a process of critical thinking and reflection. This is crucial in the context of agricultural research that deals with complex socio-technical–environmental systems. Programmes that seek to holistically approach modelling uncertainties are one such interdisciplinary research setting, and we argue that equally valued disciplinary contributions must be included already during problem framing and design stages, which should then translate into adequate resourcing and integration of different research areas. However, implementing these considerations in practice can be difficult. In the following sections, we reflect on the process and challenges that can emerge during collaborative problem framing through an investigation into uncertainties in sensor measurements and biophysical models associated with agricultural digitalization. To do so, the next section outlines three distinct disciplinary perspectives on modelling uncertainty.

# 3. The interconnected facets of modelling uncertainty

Nearly all scientific disciplines have something to say about uncertainty. It is an elusive term that, like other polysemic notions, evokes diverse meanings [29]. An initial understanding of uncertainty as a 'lack of complete knowledge' seems plausible but does not allow differentiation from associated terms such as risk or ignorance, particularly when these terms are used across multiple disciplines and contexts. This semantic ambiguity—itself linguistic uncertainty [30]—can be problematic when agricultural or environmental policy decisions are based on uncertain information. In this section, we, therefore, address the question of 'How might uncertainty manifest in the context of biophysical modelling?' through three disciplinary lenses to highlight the multitude of facets that warrant consideration.

Our intent is to demonstrate the interconnectedness of sociocultural, ecological, technical and statistical aspects in regard to modelling uncertainties. In stressing that social, biophysical and technological contexts

equally matter, we seek to make the case for systematic reflections on modelling uncertainties that do not falsely dichotomize between objective, technical aspects 'within' models and subjective, sociocultural aspects 'outside' of the scientific practices of data generation and modelling. An appreciation for the entanglements of different modelling uncertainties highlights the need for interdisciplinary research that includes collaborative problem framing and shared knowledge creation.

## 3.1. A social science lens

It is insightful to frame uncertainty within the interplay of knowing and not-knowing more broadly. As a sociocultural phenomenon that involves sociality, knowledge depends on knowing individuals who are suspended in social relationships and cultural webs of meaning [31,32]. Knowing is not only the possession of externalizable information but 'cognitive doing' by situated agents—knowledge must be known. It forms part of individuals' capacity to act upon and engagement with the biophysical and social world [33,34]. Correspondingly, not-knowing is not merely an absence of information but also 'provides the grounds for action, thought and the production of social relations' ([35, p. 21]; also [36]).

In between the extremes of 'unknown unknowns' and settled knowledge thus stretches a wide spectrum that includes known or suspected unknowns, not-yet-knowns and desired unknowns.[2] The impetus for scientific research, for instance, often emerges from partial knowledge or curiosity of the unknown [38,39]. Uncertainty can fulfil similar positive functions, from opening possibilities for creativity to generating excitement over the outcomes of sporting events [29,40]. Not-knowing is, in this sense, not necessarily negative. A relational perspective, therefore, regards unknowns in tandem with what is (claimed to be) known. The resulting epistemic complexities require clear terminologies. Figure 1 outlines a frequently cited taxonomy. While this taxonomy warrants critical evaluation, it emphasizes that uncertainty is only one manifestation of unknowns and that these should be distinguished. Moreover, uncertainty itself can be differentiated.

Uncertainty and risk in particular are two commonly used terms. Both refer to partial knowledge of unknowns, albeit to varying degrees. Here, Wynne's [42] differentiation of four unknowns in policy contexts is insightful. *Risk* implies that systems' behaviours and the odds of different outcomes are well known and quantifiable, while *uncertainty* suggests that only general system parameters but not their probability distributions are known (also [43,44]; [45, p. 31]; cf. [46]). *Ignorance* is a third state where 'we don't know what we don't know', which is relevant when decisions are made without knowledge of, for example, unintended consequences. *Indeterminacy* highlights the contingencies associated with human behaviour. Even if high scientific certainty exists, policy contexts are often characterized by a 'real open-endedness in the sense that outcomes depend on how immediate actors will behave' ([42, p. 117]; cf. [47]). Beck [48, p. 291] equally stresses the distinction between 'specific calculable uncertainties—"risks"—which are determinable with actuarial precision in terms of a probability calculus' and many contemporary hazards that manifest as *manufactured uncertainties*. These are 'dependent on human decisions, created by society itself, immanent to society and thus non-externalizable, collectively imposed and thus individually unavoidable' [48]. Questions of predictability can thus affect uncertainties as well as the role of modelled information in decision- and policy-making.

We would contend that some uncertainties are quantifiable and relevant probability distributions can be known (e.g. [10,49]). Risks are, therefore, not the only type of quantifiable unknowns, nor are all risks quantifiable, especially if viewed through a cultural theory lens [50]. Likewise, ignorance may not only refer to 'unknown unknowns' if understood as a more dynamic sociocultural phenomenon [51, pp. 30–39]. The outlined distinctions nonetheless highlight that unknowns should be differentiated carefully and that knowledge of unknowns, including uncertainty, is often *conditional* and *context-specific*; almost certain outcomes in one setting may be uncertain in another due to altered parameters, and may depend upon assumptions about human or non-human behaviour.

Applying such a relational perspective to environmental management shows that '[u]ncertainty refers to the situation in which there is not a unique and complete understanding of the system to be managed' [52, p. 4]. Attending to uncertainty's equivocality suggests that in addition to *ontological/aleatoric* uncertainty (e.g. unpredictability or randomness) and *epistemic* uncertainty (e.g. lack of settled knowledge), a third kind of uncertainty exists: *ambiguity* can emerge from multiple knowledge frames through which uncertainty gains meaning ([52,53]; also [43,54]). Uncertainties must thus be considered in the context of actors' social and material relationships, within which it might be

---

[2]Wehling [37, pp. 116–149] examines three interconnected dimensions for characterizing unknowns: the level of knowledge about unknowns, the intentionality of not-knowing and the temporal stability of unknowns.

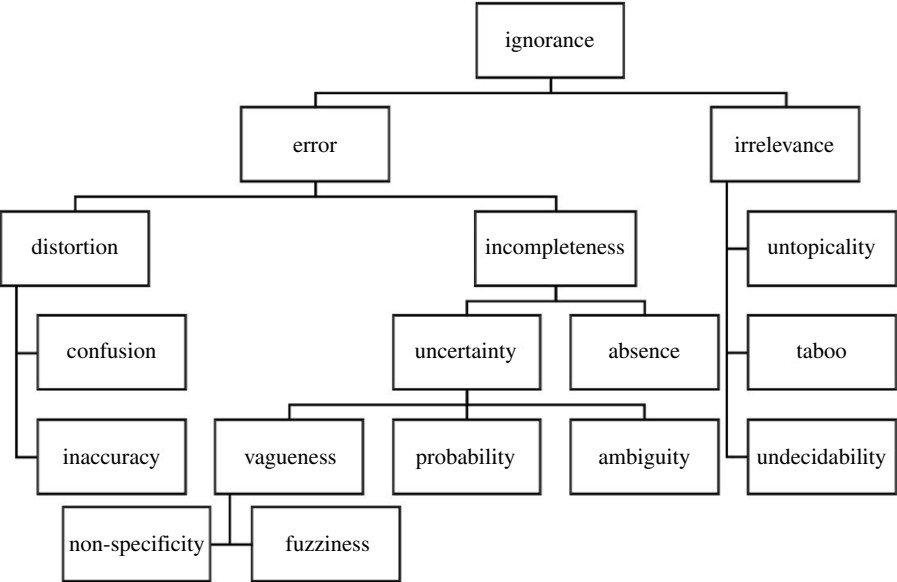

**Figure 1.** Taxonomy of ignorance (source: authors based on [41]).

contested, strategically employed, or systematically excluded.[3] 'More and better' scientific research can resolve some ambiguities, but might, conversely, also lead to a heightened awareness of uncertainties [39,55–57]. Smithson [29], therefore, contends that uncertainties are socially constructed. We tentatively concur that uncertainties are socially *co*-constructed in tandem with biophysical and technological aspects. Combining symbolic and material characteristics means that 'uncertainty cannot be understood in isolation but only in the context of the socio-technical–environmental system in which it is identified' [52, pp. 11–12]. It is thus crucial to equally reflect on technical, environmental and sociocultural aspects of uncertainty.

This perspective has implications for the types of uncertainties associated with environmental and biophysical models. These models are employed in various ways, which makes it difficult to generalize related uncertainties. Models are used to test hypotheses; describe and explain data patterns; synthesize disparate data; integrate, impute and evaluate data; or to predict future systems behaviours [5,6,58,59]. All these functions are accompanied by uncertainties. Ecosystem models, for example, never fully encompass the systems that they represent in simplified form [60,61]. These problems are compounded if quantitative models are used deterministically to make longer-term predictions [45,60–64]. Another factor is how modellers produce and communicate modelled information. In negotiating the validity of their knowledge claims with other modellers or actors, modellers can strategically mobilize, underplay or selectively acknowledge known aspects leading to uncertainties, while remaining unaware of others [65–67]. Modelling uncertainties thus not only manifest due to biophysical system properties but also sociocultural factors associated with the production and dissemination of modelling outputs.

This performativity of modelling and the communication of modelled information are crucial influences on how modelling uncertainties are understood by different actors. This has also been acknowledged outside of the social sciences. The statistician and journalist Silver [45, p. 14], for instance, highlights the performative character of modelled predictions and use of 'big data' by noting that 'numbers have no way of speaking for themselves. We speak for them. We imbue them with meaning'. Sociocultural perspectives on modelling uncertainty can, therefore, not only inform an understanding of the social engagements with modelling outputs, but also provide insights into the processes of model building themselves by detailing the context and underlying conditions of their production.

## 3.2. A modelling lens

The understanding and management of uncertainties is a growing research area within the modelling community, with the distinct characteristics between types of models and applications requiring

---

[3]On the intentionality of not-knowing, see Espig [51, p. 36].

different priorities in uncertainty management. Here, too, uncertainties can manifest in diverse facets. Where models support decision-making, for instance, Walker *et al.* ([54, p. 8], original emphasis) generally define uncertainty 'as being *any departure from the unachievable ideal of complete determinism*' and propose three distinct dimensions of differentiation around the location, level and nature of uncertainty (cf. [11]). Walker *et al.* [54] describe uncertainty as either epistemic or variable to highlight that uncertainties emerge from imperfect knowledge of socioecological systems as well as inherent variability and unpredictability within these systems—that is, ontological or aleatory uncertainties. While ontological uncertainties are perceived as largely irreducible, epistemic uncertainties are associated with incomplete knowledge, inaccurate data or imperfect models. Precisely differentiating reducible and irreducible modelling uncertainties is, therefore, important.

The need to systematically understand uncertainties is increasingly recognized within the modelling community and has led to critical assessments of models' performance, for instance, where biophysical models support agricultural regulation. Four terms are often used almost interchangeably in such evaluations: accuracy, error, precision and uncertainty. Shepherd *et al.* [68, pp. 3–4] clarify differences between these terms. *Accuracy* refers to the 'degree of closeness of measurements of a quantity to that quantity's actual (true) or accepted value', while *error* describes the 'difference between the modelled representation of a system, and the reality of the system'. Input errors emerge from measurement mistakes around parameters such as soil or weather data. Model errors are faults in a model itself due to, for instance, conceptual misunderstandings of the modelled system or incorrect model calibration and validation. Output errors are the result of both input and model errors. *Precision* is the 'degree to which repeated measurements under unchanged conditions show the same results'. It is, therefore, also described as reproducibility or repeatability. *Uncertainty* is perhaps the least clearly defined term and relates to 'a potential limitation in some part of the modelling process that is a result of incomplete knowledge' [68]. Based on Walker *et al.* [54], Shepherd *et al.* [68] highlight five sources of modelling uncertainty: context and framing, inputs, model structures, parameters and model implementation. The authors apply these insights to evaluate the performance of a biophysical model that is used as a regulatory instrument to predict nutrient loss on New Zealand farms (cf. [13]).

Bammer [58, p. 35] contends that most advances in dealing with uncertainty have been made around modelling, including the propagation of uncertainties in complex systems or future scenarios (cf. [62]). Various methods exist to address probabilistic and statistical uncertainties, vagueness or imprecision. For instance, the development of methods to quantify uncertainty in deterministic biophysical models is an active area of research and many tools are in common use, ranging from statistical objective functions, multi-model ensembles, sensitivity analysis and emulators [7,10,49,69]. Meenken *et al.* [9] contend that a systematic approach to identifying biophysical modelling uncertainties includes understanding the environment in which a model operates, and that the sources and reasons for uncertainty are identified. To do so, they propose a formalized framework for quantitative uncertainty evaluation. The authors argue that identifying and discussing components of biophysical modelling uncertainty through this framework provides an opportunity to target resources to reduce the overall size of uncertainty.

While often focused on the quantitative uncertainties *in* models, this growing body of research signals that differentiating uncertainty from closely related notions is indispensable for robust understandings and communication of models' performance. Schmolke *et al.* [5, p. 481] note that, in the context of ecological models, '[i]nconsistent and controversial terminology … hampers the establishment of good modeling practice'. It has also been realized that modelling outputs cannot be assessed in isolation, especially where they support decision- or policy-making. It is, therefore, pivotal to not only develop clear terminologies, but equally assess if models adequately correspond to an intended outcome and whether suitable modelling steps are implemented [54,70–72]. In practice, however, it can be difficult to attend to the myriad quantitative and qualitative uncertainties that might affect models' (perceived) performance, especially for individual researchers or disciplinary-bound teams. Interdisciplinary research projects can enable more holistic and robust approaches to understanding modelling uncertainties. This requires shared terminologies and epistemic frames, as discussed in §§4 and 5.

## 3.3. An engineering and data science lens

A third area of research within our interdisciplinary team concerns measurement and data uncertainties. On-farm and on-animal sensors are key components of digitally enabled agricultural systems. The likelihood of undesirable decision outcomes within emerging smart farming systems can only be

managed when the accuracy of available information is known. Questions around measurement error are therefore highly pertinent for considerations around modelling uncertainties. As Oreskes [60, p. 16] points out:

> in order to make observations in the natural world, we invariably use some form of equipment and instrumentation. … [O]ur tests have become progressively more complex, and apparent failures of theory may well be failures of equipment or failures on our part to understand the limitations of our equipment.

The relationship between data and the model depends on the application. To be considered robust for the functions described above, biophysical models should generally be calibrated and validated using data that are derived from observation or measurement equipment. Further, many models will also require context-specific input data in many research and decision-making applications. Within digital agricultural systems, such measurement data are increasingly generated through multiple connected sensors, Internet-of-Things devices and information that is itself derived from data models. Additionally, the understanding of data quality and its completeness can be inadequate, when, for example, disparate datasets are combined in machine learning applications [73]. As the volume and variety of digital data increases, so too do associated uncertainties. Understanding modelling uncertainties should thus involve consideration of measurement and data uncertainties.

Doing so, once again, requires clear terminology. Measurements, for instance, are always mere approximations since all measurements are subject to error [74]. However, measurement errors are not synonymous with measurement uncertainty [75,76]. *Error* defines the difference between the exact value of a quantity of interest (the measurand) and a measured value.[4] The error is never known, but it is possible to know the range of error values that can occur. These errors can be either random (aleatory) or systematic. *Random errors*, so-called noise, can arise from unpredictable spatial or temporal changes that influence data generation and are thus context- and time-specific. *Systematic errors*, sometimes called 'bias', are enduring and often consistent across different observations. Systematic errors are frequently a residual offset, or a scale factor, that can be corrected to some degree through the calibration of a measurement system [76]. Such errors can lead to poor sensor data quality and reduce the usefulness of that data in decision-making processes [77].

The *uncertainty* associated with measurement data represents doubt about how close the measurement is to the exact value of the quantity of interest [75,76]. This doubt is expressed by taking a measurement result as an approximation for the measurand and may be thought of as expressing the quality of the measurement procedure. Assessing measurement procedures requires further differentiation between precision and accuracy [74–76]. *Accuracy* describes how close a measurement is to an accepted value, which can be affected by systematic errors that are introduced through both measuring instruments and measuring procedures. *Precision* signals how close individual values are to each other and indicates the repeatability of a measurement. Precision of measured data is influenced by random errors that may cause data to fluctuate around a mean value and this fluctuation contributes to uncertainty. Sharifi *et al*. [78] detail how measurement uncertainty can manifest in an animal activity and behavioural sensor system. Based on this analysis, they contend that uncertainties will inevitably occur due to several sources of error, and stress that no statement of measurement results is complete without an evaluation of resulting measurement uncertainty. The authors propose that developing a robust measurement uncertainty framework can help to identify, evaluate and visualize these uncertainties.

This brief overview shows that data generation is not merely the collection of biophysical facts but already a performative process that can introduce uncertainties unrelated to the inherent variability or randomness of biophysical processes, for example, due to false assumptions [61, p. 642]. Notions such as 'raw data' can thus be misleading if the context and process of data generation are neglected ([51, pp. 174–181]; [79,80]). The sociocultural considerations we outlined above, therefore, not only concern contexts 'outside' the scientific practices of data generation and modelling, but equally form an integral part of these practices, with relevant implications for uncertainty in and around modelling. One immediate implication is that generating more data, as in the context of agricultural digitalization, does not automatically lead to less uncertainty—indeed, the growing usage of digital data might increase modelling uncertainties or introduce new forms of uncertainty.

---

[4]We here refer to the exact value from a particular metrological perspective [74]. Under a Bayesian paradigm, for example, one would not look for an exact value but an underlying distribution.

Combining the insights from these three perspectives demonstrates that the complexities and subtleties of biophysical modelling uncertainty in digital data-rich contexts cannot be reduced to single numbers or quantitative assessments alone (also [81]). Systematic reflections on modelling uncertainties should thus consider sociocultural, technical, ecological, biophysical and statistical aspects. Understanding uncertainties *in and around* biophysical modelling holistically is not only of scholarly importance but can help to address challenges regarding models' legitimacy in decision-making processes and legal implications [5,13,82,83]. This includes framing and communicating such understandings effectively. Otherwise, decision-makers and other users may be less or over-confident about modelled information, which can lead to poor or detrimental outcomes [11,12,84–86]. Our relational perspective on modelling uncertainties, therefore, stresses that *social, biophysical and technical contexts matter*.[5]

## 4. Conceptualizing uncertainties in and around modelling

It is, prima facie, difficult to determine how to holistically consider the described facets of modelling uncertainty. Helpful classification systems have been proposed (e.g. [54,85]; for a systematic review, see [11]), but these are less applicable in the initial stages of interdisciplinary research due to their specificity, which requires existing shared epistemic frames. They also do not fully account for the contextual relationships of socio-technical–environmental systems. In a first step to overcome these challenges, we developed a simple conceptual framework[6] of different meta-categories of uncertainties in and around modelling. Our framework is intended as a starting point for interdisciplinary projects and a broader understanding of modelling uncertainties. This section introduces the tenets of this framework and we reflect on its usefulness in the next segment.

Our conceptual distinctions build on van der Bles *et al.* [86] who suggest two different levels of uncertainty.[7] They distinguish between uncertainty directly about epistemic 'objects' and more indirect 'meta-uncertainty' about the underlying evidence for their assessment [86, p. 7]:

> *Direct* uncertainty about the fact, number or scientific hypothesis. This can be communicated either in absolute quantitative terms, say a probability distribution or confidence interval, or expressed relative to alternatives, such as likelihood ratios, or given an approximate quantitative form, verbal summary and so on.

> *Indirect* uncertainty in terms of the quality of the underlying knowledge that forms a basis for any claims about the fact, number or hypothesis. This will generally be communicated as a list of caveats about the underlying sources of evidence, possibly amalgamated into a qualitative or ordered categorical scale.

Van der Bles *et al.* provide a fitting legal analogy to illustrate the distinction between direct and indirect uncertainty (see box 1). However, their article primarily examines the communication of direct uncertainty. Understanding direct uncertainties in the first place requires insights into how and why a specific dataset or model were produced [6,71,82]. For instance, modellers often choose data and models not only due to best scientific suitability but also practical limitations, such as access to specific datasets or availability of specific modelling software. We, therefore, regard data generation, model selection and modelling as active processes that already involve assumptions and decisions, which can introduce direct uncertainties and discrepancies between an intended outcome and chosen data or model [54]. Going further, we find van der Bles *et al.*'s differentiation insightful for considering not just uncertainties in, but also around, modelling. The latter, however, necessitates expanding upon the direct–indirect distinction.

Following a relational perspective, we integrate social insights into van der Bles *et al.*'s distinction, firstly, by adding social actors to the understanding of indirect uncertainty. Our addition suggests not only a concern with underlying knowledge *per se* but includes those who negotiate that knowledge; that is, we incorporate actors and their sociality. This is salient in cases where

[5]See also postconstructivist [87] and critical realist [88] approaches, which attempt to transcend the nature–culture dichotomy. For a corresponding anthropological perspective, see Gingrich [23].

[6]We share Brondizio's [16] caution about referring to a 'conceptual framework', which, as a term, is 'polysemic and evokes different images in people's minds, and as such it can easily derail the conversation into unproductive domains'. We agree that, in the context of socioenvironmental research, 'a conceptual framework refers to articulating the "big picture" in a way that brings together the multiple parts of a puzzle and their interrelationships. It is a type of system thinking that can be expressed in narrative … or schematic ways' [16].

[7]Van der Bles *et al.*'s [86] conceptualization of 'levels' of uncertainty is somewhat different to Walker *et al.* [54], who propose it as one of three dimensions to differentiate uncertainties (see §3.2). We here follow van der Bles and co-authors when referring to levels of uncertainty.

**Box 1.** The levels of uncertainty in legal reasoning: a relationally expanded analogy.

Van der Bles *et al.* [86, p. 8] present two levels of uncertainty related to the archetypical components of a legal case that seeks to determine a suspect's guilt:

'*Direct uncertainty* concerns the absolute probability of guilt, and the relative "probative value" given to an item of evidence for or against guilt of this particular suspect'. Communicated verbally, 'direct absolute uncertainty may be expressed as "beyond reasonable doubt", direct relative uncertainty may be communicated by saying some forensic evidence "supports" or "is consistent with" the guilt of the accused'.

'*Indirect uncertainty* would be reflected in the credibility to be given to an individual's testimony concerning this item of evidence. … The indirect quality of the background knowledge might be introduced in cross-examination by querying the competence of the forensic expert or their access to appropriate data'.

Indirect uncertainty can be relationally expanded to the case's sociocultural dimensions. For instance, observers or judges might indirectly evaluate presented evidence by assessing the forensic expert's qualifications, conflicts of interest and trustworthiness—thus judging the direct uncertainty by proxy. This may be the only way for non-experts to assess the evidence's 'scientific certainty'.

The setting and actors' relationships are also important. For example, observers may trust the forensic expert's ability and presented evidence, yet still be uncertain about how judges will incorporate this testimony, especially when multiple, potentially conflicting or incommensurable sources of evidence are considered. The court might also be under external pressure to deliver a certain verdict, which can add uncertainties beyond those directly or indirectly associated with the evidence.

*Contextual uncertainty* can thus emerge from the wider court setting and actors' positions within, and experiences of, the legal process. Contextual factors may manifest in doubts over the procedural fairness of court proceedings.

uncertainties are negotiated around socioenvironmental issues, for instance, when biophysical models are used in the regulation of agricultural systems [13]. Within these complex settings, fully assessing specific facts or the quality of underlying knowledge might only be feasible for expert practitioners or researchers. It is, therefore, reasonable for non-experts to assess direct and indirect uncertainties by assessing the qualifications, motives or trustworthiness of those presenting modelled information and uncertainty evaluations—an assessment 'by proxy' so to speak. This facet of indirect uncertainty probably manifests across most contemporary contexts where models are used as decision-support tools.

Secondly, uncertainties are conditional, context-specific and potentially subject to multiple epistemic frames. We thus propose a complementary third level of uncertainty:

*Contextual* uncertainty about the wider setting of the production and application of facts, numbers or scientific hypotheses. Similar to indirect uncertainty, it can be expressed qualitatively but relates to the background conditions within which facts, number or scientific hypotheses are produced and become embedded once they are utilized for decision- or policy-making.

Benessia & De Marchi [89] similarly refer to 'situational uncertainty' in the aftermath of the devastating 2009 earthquake near the Italian city of L'Aquila. They outline multidimensional uncertainties that cannot be reduced to scientific factors but also emerge from legal, moral, societal, institutional and proprietary aspects associated with natural disaster decision-making and uncertainty management. In the context of biophysical modelling in the agricultural sector, contextual uncertainty may manifest for some actors if it is unclear how modelled information will be used in environmental regulation, regardless of direct and indirect uncertainties associated with the corresponding model (e.g. [13]). Attending to models' background settings thus problematizes a reduction of complex sociocultural phenomena solely into scientific conceptions ([90, p. 84]; [91]). At the same time, considering contextual factors *in tandem with* direct and indirect uncertainties avoids a misleading bifurcation into scientific and non-scientific aspects. By placing emphasis on their interrelatedness in actual settings, we thus stress the need for holistic uncertainty analyses that require combining specific disciplinary insights and the integration of multiple epistemic and ontological frames.

Box 1 provides a legal analogy of the three-level framework we propose.[8] Such an analogy is equally applicable to agricultural regulation where policymakers frequently decide regulatory measures based on uncertain modelled information and myriad other contextual factors—all of which can introduce diverse uncertainties for policymakers, modellers and farmers. While the example in box 1 illustrates our three-level conceptual distinction, two points warrant clarification before reflecting on the usefulness of this framework in interdisciplinary research. First, we do not suggest that modelled information becomes quantitatively or qualitatively more uncertain once incorporated into decision-making contexts. Rather, the aspects that warrant consideration widen. This means, for example, that direct uncertainties within a model might be underplayed or overstressed by actors who seek a specific policy outcome within a given context [65,66]. Presented modelled information can then appear more or less certain, as judged against technical or statistical criteria. Alternatively, particular direct or indirect uncertainties may be deemed irrelevant within a specific decision-making scenario, which is why Doyle *et al*. [11] suggest a focus on *decision-relevant uncertainty*. Seen holistically, uncertainties are, therefore, co-constructed by social, biophysical and technological aspects.

Second, modelling practices and the use of modelled information also do not progress linearly through separate stages, but usually involve iterative steps and, ideally, feedback loops between data generation, modelling and application. The conceptual distinction between direct, indirect and contextual uncertainties does, therefore, not describe separate stages or unidirectional flows of data and modelled information between them. In this sense, contextual factors such as political or funding priorities do already significantly influence the processes of data generation and modelling.

# 5. Reflection: ordering interdisciplinary research through collaborative problem framing

A desire for disciplinary complementarity is a prerequisite for successful interdisciplinary projects, but establishing meaningful collaboration can be challenging. Bammer's work on Integration and Implementation Sciences (I2S) is insightful. She highlights crucial core principles that should guide interdisciplinary research, including systems-based thinking, shared problem framing and boundary setting, openly addressing values, and collaboration [58,93]. Synthesizing the knowledge from researchers and stakeholders with different disciplinary backgrounds can occur through three overarching methods. *Dialogue-based* methods aim to co-create meaning and shared understandings between participants [93]. *Model-, product- or vision-based* methods are focused on establishing a commonly shared goal or object of study. Here, a model can be 'a device which provides a focal point for discussion and action between people representing different disciplinary perspectives and different types of practical experience … . It provides a way of organizing different pieces of information' [58, p. 34]. *Common metric-based methods* can generate synthesis through a single measure that encapsulates shared values, for instance, monetary value in cost–benefit analyses [58,93]. Bammer's I2S approach outlines further guiding elements, but it is the notion of synthesizing multiple epistemic frames to harness the potential of explanatory pluralism that is helpful for reflecting on the usefulness of our conceptual framework.

A lack of shared uncertainty definitions initially constituted a major challenge for our team. While the notion of 'modelling uncertainty' sparked a common interest, this was probably due to the term's polysemic ambiguities, which makes it ostensibly inclusive but too vague to generate shared understandings of the different facets of uncertainty. As outlined in §3, those of us working on engineering projects referred to uncertainties from sensor measurement error or equipment failure. The team's social scientists highlighted that conflicting understandings of modelled information or a lack of trust could equally contribute towards modelling uncertainties. Consequently, we spoke near-different languages regarding underlying meanings of uncertainty. External collaborators, therefore, recommended not to use the term uncertainty at all. These semantic difficulties highlight the multiple knowledge frames through which uncertainty can gain meaning [52], which can hamper interdisciplinary research efforts.

Our conceptual framework helped to clarify some of this confusion and formed the basis for more meaningful dialogue and system-based thinking. Specifically, the outlined meta-categories of direct, indirect and contextual uncertainty allowed us to move towards a shared appreciation of the

---

[8]The conceptual framework can also be represented schematically (see [92]).

different facets of modelling uncertainties that our individual research projects addressed. Realizing the complementarity between our approaches led to a better understanding of the aspects of biophysical modelling that might be uncertain, enabled us to systematically locate corresponding individual contributions and created a mutually shared recognition of disciplinary strengths and limitations [16,94]. For instance, improved measurement accuracy might be insufficient for resolving indirect uncertainties emerging from farmers' mistrust of modellers or of a specific dataset, and vice versa. Framing different uncertainties across projects has been crucial in delineating work areas and establishing productive collaboration, as evidenced by a joint panel on *Uncertainty in Measurement and Modelling* at a New Zealand-based agricultural management conference [95]. In this sense, the *process* of co-designing the three-level conceptual distinctions—perhaps more so than the framework itself—contributed to the creation of discursive and interactive spaces to explore knowledge production beyond disciplinary boundaries, which involved building a trusted team culture that allowed for constructive dialogue. Regarding interdisciplinarity, in the first instance, as collaborative problem framing and the opening up of spaces for critical reflection is thus crucial [16,58].

From this perspective, our framework enabled us to synthesize multiple epistemic frames into an interdisciplinary approach that has more explanatory potential than discipline-specific insights by themselves. The framework here functions as an interdisciplinary 'boundary object' that is 'both plastic enough to adapt to local needs and constraints of the several parties employing [it], yet robust enough to maintain a common identity across sites' ([96, p. 393]; also [18,97,98]). Crucially, developing the framework allowed for a first-order delineation of different levels of modelling uncertainty, thereby bridging disciplinary divides and opening opportunities for meaningful collaboration. Our approach thus contributed towards ordering the mosaic of different disciplinary and individual perspectives within our team. This suggests that co-developing conceptual frameworks, narratives or schemata can help to initiate effective interdisciplinary research more broadly.

Nonetheless, such processes include obstacles and hurdles. Turning a desire for disciplinary collaboration into productive interdisciplinary research thus requires deeper reflection on how diverse ontological, epistemological and methodological positions can be brought together equitably and in ways that establish complementary insights. As Brondizio [16] notes:

> The challenge is to articulate ways to leverage this complementarity as part of a larger puzzle, including the contradictions that emerge from diverse theoretical orientations, knowledge systems, and types of questions and evidence.

Conceptual frameworks and joint narratives can provide a common direction for interdisciplinary research by opening conversational spaces for collaboration and generating creative disciplinary tensions [94,99]. While such boundary objects are important for initiating interdisciplinarity, they do not, however, necessarily resolve institutional tensions and inequalities in terms of disciplinary relations. Verma *et al.* [22, p. 257] highlight the 'negotiation, bargaining and, sometimes, contestation and resistance between and among different domains of disciplinary actors, knowledge, meanings and understanding' in the context of biophysically dominated agricultural research. As our team equally experienced, successful interdisciplinary research cannot easily be established on a single programme basis with limited timeframes and funding, as it initially requires significant time to build relationships and mutually shared understandings. Within shorter-term projects, it can be difficult to have such capacity building institutionally recognized and counted towards researchers' traditional performance criteria (e.g. [91]). It is, therefore, crucial to reflect on these challenges and the underlying logics for interdisciplinarity.

Moving beyond limited functional understandings of ad hoc interdisciplinary research requires a more fundamental reconfiguration of the relationship between the natural, social and applied sciences. This should include re-evaluating unequal disciplinary standings that can, for example, lead to fewer funding opportunities for research on the sociocultural facets of modelling uncertainty and agricultural digitalization when compared with their biophysical aspects. Realizing such an *ontological* logic of interdisciplinarity places new demands and responsibilities on agricultural research organizations, funding agencies and researchers. This includes not merely engaging in research that is framed as interdisciplinary, but to foster the 'cultivation of interdisciplinary subjectivities and skills' [21, p. 39] that embed interdisciplinarity at the core of institutional policies and everyday research practices. We propose that collaborative problem framing can contribute towards this end, may it be as a starting point for exploring modelling uncertainties [71,98] or as the foundation for holistic research on pressing environmental challenges more generally [16,26,58,93,99].

# 6. Concluding remarks

We developed two complementary insights in this article, both around the object and the process of investigating uncertainty in agricultural decision-making and regulation that is increasingly driven by information derived from digital data sources. First, we outlined the diverse quantitative and qualitative uncertainties associated with sensor measurements and biophysical models in these contexts. This led to the argument that uncertainties must be analysed within the socio-technical–environmental systems in which they are identified. In other words, social, biophysical and technical contexts all matter for holistically understanding modelling uncertainties. To capture these insights, we introduced a conceptual framework that is based on a three-level distinction between models' direct, indirect and contextual uncertainties, which allows for first-order delineations of different modelling uncertainties. Investigating these interrelated uncertainties requires diverse disciplinary approaches that are grounded in a range of ontological, epistemological and methodological perspectives.

Second, we embedded this argumentation into a reflection on interdisciplinary research within biophysically dominated agricultural research organizations. The process of co-developing the proposed conceptual framework opened opportunities for meaningful dialogue and productive collaboration, with the framework functioning as a boundary object that helped to bridge disciplinary divides. However, we highlighted some of the challenges of facilitating successful interdisciplinarity and cautioned against merely employing interdisciplinary research on an ad hoc basis to address pre-defined research questions. Instead, we contend that a deeper engagement with interdisciplinarity should occur both among individual researchers and on an organizational level, which prompts a more fundamental reconfiguration of the relationships between the social, natural and applied sciences. This involves collaborative problem framing and ongoing reflection on disciplinary strengths and limitations. Such approaches are particularly salient in the analysis of modelling uncertainties associated with complex socio-technical–environmental systems.

Interdisciplinary research is, however, only one—perhaps initial—response to the intricacies of contemporary socioenvironmental challenges, which raises questions about the linear logic of disciplinary knowledge production and its institutional structures. Mounting sustainability challenges, for instance, demand approaches that do not reduce the interconnected social, technical and biophysical aspects of 'real-world problems' into isolated parts but, instead, start by accepting these complexities as part of holistic, problem-oriented research. Interdisciplinary integration within a research organization can help to address such problems, but it may still lack practical engagements with the local knowledges and lived experiences of land managers, regulators and policymakers, or local communities. Transdisciplinary approaches are, therefore, required that purposefully include 'non-academic' or 'non-scientific' participants [15,91,100]. In this sense, responsible research should ultimately lead to collaborative problem framing not only among researchers from multiple scientific disciplines but a variety of actors, and their diverse forms of knowledge. In the context of simulation modelling, such transdisciplinary approaches are particularly important when modelled information is intended to contribute meaningfully to the development of policy pathways ([98]; cf. [91]). Our experiences regarding agricultural digitalization and associated modelling uncertainties support this view.

The proposed conceptual framework can be a productive starting point for discussion among a wide range of actors by providing a first-order distinction between uncertainties in and around biophysical models. Discerning the diverse quantitative and qualitative modelling uncertainties in the context of agricultural decision-making and regulation can be difficult (e.g. [13]). Effectively addressing them is nonetheless crucial for realizing the full potential and societal benefits of agricultural digitalization [1,8]. Inter- and transdisciplinary approaches are promising, but they can be hampered by practical and conceptual challenges, including entrenched attitudes, biases and institutionalized disciplinary inequalities. These challenges might perpetuate epistemic fragmentation and limit the establishment of collaborative spaces. Ongoing critical reflection on the likely messy, nonlinear and laborious processes of collaborative knowledge production is, therefore, indispensable.

Ethics. This article does not contain primary, or otherwise empirical, data, apart from the authors' insights and reflections. Human research conducted as part of a bigger programme, but which is not included into this article, was approved under AgResearch's Social Research Ethics process (Ethics Approval No. 9/2019).
Data accessibility. No primary or empirical data are included in this article, apart from the authors' insights and reflections.

**Authors' contributions.** All authors substantially contributed to the conception and design of the manuscript. All authors approve the manuscript's final version and agree to be accountable for all aspects of the work. M.E. principally drafted and revised the manuscript. M.E. developed §§1, 3.1, 4 and 6 with substantial intellectual contributions and review from each co-author. S.C.F.-S. and M.E. co-developed the argumentation of §§2 and 5, with input and review by E.D.M., D.M.W. and M.S. S.C.F.-S. further provided substantial input into the overall conception and reviewed several draft versions. E.D.M. and D.M.W. provided material for §3.2, which was adapted for the manuscript by M.E. E.D.M. and D.M.W. further provided substantial input into the overall conception and reviewed several draft versions. M.S. provided material for §3.3, which was adapted for the manuscript by M.E. M.S. further provided substantial input into the overall conception and reviewed several draft versions.

**Competing interests.** We declare we have no competing interests.

**Funding.** This research is part of AgResearch's *New Zealand's Bioeconomy in the Digital Age* (NZBIDA) platform, funded by the New Zealand Ministry of Business, Innovation & Employment.

**Acknowledgements.** We are grateful for critical pre-submission reviews by our colleagues Dr Alasdair Noble and Dr Margaret Brown. Dr Thomas Wright also provided valuable feedback. We thank two anonymous reviewers for their insightful comments.

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
