## [Reviewer comments · Royal Society Open Science]

Review History

RSOS-201511.R0 (Original submission)

Review form: Reviewer 1

Is the manuscript scientifically sound in its present form?

Yes

Are the interpretations and conclusions justified by the results?

Yes

Is the language acceptable?

Yes

Do you have any ethical concerns with this paper?

No

Have you any concerns about statistical analyses in this paper?

No

Recommendation?

Major revision is needed (please make suggestions in comments)

Comments to the Author(s)

File attached (Appendix A).

Review form: Reviewer 2**Is the manuscript scientifically sound in its present form?**

No

Are the interpretations and conclusions justified by the results?

No

Is the language acceptable?

Yes

Do you have any ethical concerns with this paper?

No

Have you any concerns about statistical analyses in this paper?

No

Recommendation?

Major revision is needed (please make suggestions in comments)

Comments to the Author(s)

The paper reports on insights regarding uncertainty in and surrounding biophysical modeling. These insights were gained during an interdisciplinary research project on agricultural digitalization. I struggle with giving a recommendation on this manuscript. On the one hand, parts of the manuscript are insightful and contain useful distinctions and discussions. On the other hand, I fail to articulate what the main contribution is that the authors are trying to make. Moreover, I find large parts which dwell on interdisciplinarity tedious. If the authors can articulate more clearly in the introduction what the gap in the literature is that they are trying to address, and streamline the remainder of the manuscript around this contribution, I believe this will be a relevant paper worthy of publication. Let me elaborate this in more detail

First, I completely miss the point of section 2. How is section 2 adding to the contribution the paper is trying to make? In part, the issue might simply be that I lack a clear disciplinary identity having been trained in and always having worked in interdisciplinary research. However, the issue I have with this section is also that it is not clearly connected to the focus of the paper: uncertainty in and surrounding biophysical modelling.

Second, on page 8, you correctly highlight that ambiguity is a kind of uncertainty next to epistemic and ontic (not ontological, but this is a fine philosophical point) uncertainty. This claim is not novel. Kwakkel, Walker, and Marchau (2010), with reference to Dewulf, Craps, Bouwen, Taillieu, and Pahl-Wostl (2005), explicitly include ambiguity within the nature dimension of Walker et al. (2003). However, since ambiguity is about knowing differently, it is an epistemic notion. Thus, the correct way is to make a distinction first between uncertainty as an attribute of the world (i.e. ontic or aleatoric uncertainty) and uncertainty due to our knowledge about the world (i.e. epistemic uncertainty). Within epistemic uncertainty, then, a further distinction can be made between not knowing enough and knowing different (i.e. ambiguity). As an aside, the nature dimension as conceptualized by Walker et al. originates in the first edition of Hacking (2006).

Third, the distinction between direct, indirect and contextual uncertainty is useful. However, I find it deeply confusing to call this different levels of uncertainty. Walker et al. (2003) already has a level dimension which refers to the degree or severity of uncertainty. To appropriate the level notion for something quite different, and closer to the idea of location in the Walker et al terminology, is confusing.

Fourth, the ideas developed in section 5, paragraph 3, are interesting. Walker et al have primarily used the level of uncertainty dimension as an organizing principle for the selection of ways of dealing with uncertainty. Correctly, the authors point out that the nature of uncertainty is also relevant. As pointed out by Dewulf repeatedly, if uncertainty is due to ambiguity more research will make controversies worse (Sarewitz, 2004). So, in case of ambiguity, collaborative sense making and joint fact finding are more appropriate.

Fifth, a recurring issue is that the paper interweaves two storylines. One on uncertainty in and surrounding biophysical modelling. The other containing primarily personal reflections on interdisciplinary research. The first is interesting and relevant. The second, I am unsure about its broader scientific relevance. Moreover, the interweaving of these two storylines makes the paper very dense in places and hard to follow.

Minor comment

p. 8 Line 40 "similar to brugnach et al", this statement is strange since Walker et al is older and if memory serves me well is being cited by Brugnach et al. So the logical order is that brugnach, following Wallker et al, is making a distinction between epistemic and ontic uncertainty.

Dewulf, A., Craps, M., Bouwen, R., Taillieu, T., & Pahl-Wostl, C. (2005). Integrated Management of Natural Resources: Dealing with Ambiguous Issues, Multiple Actors and Diverging Frames. *Water Science & Technology*, 52(6), 115-124.

Hacking, I. (2006). *The emergence of probability: a philosophical study of early ideas about probability, induction, and statistical inference* (2nd ed.). New York, NY: Cambridge University Press.

Kwakkel, J. H., Walker, W. E., & Marchau, V. A. W. J. (2010). Classifying and communicating uncertainties in model-based policy analysis. *International Journal of Technology, Policy and Management*, 10(4), 299-315. doi:10.1504/IJTPM.2010.036918

Sarewitz, D. (2004). How science makes environmental controversies worse. *Environmental Science & Policy*, 7, 385-403. doi:10.1016/j.envsci.2004.06.001

Walker, W. E., Harremoës, J., Rotmans, J. P., Van der Sluijs, J. P., Van Asselt, M. B. A., Janssen, P. H. M., & Kraayer von Krauss, M. P. (2003). Defining Uncertainty: A Conceptual Basis for Uncertainty Management in Model-Based Decision Support. *Integrated Assessment*, 4(1), 5-17.

Decision letter (RSOS-201511.R0)

Dear Dr Espig

The Editors assigned to your paper RSOS-201511 "Uncertainty in and around biophysical modelling: Insights from interdisciplinary research on agricultural digitalisation" have now received comments from reviewers and would like you to revise the paper in accordance with the reviewer comments and any comments from the Editors. Please note this decision does not guarantee eventual acceptance.

Please submit your revised manuscript and required files (see below) no later than 21 days from today's (ie 29-Oct-2020) date. Note: the ScholarOne system will 'lock' if submission of the revision is attempted 21 or more days after the deadline. If you do not think you will be able to meet this deadline please contact the editorial office immediately.

on behalf of Dr Agnieszka Latawiec (Associate Editor) and Pete Smith (Subject Editor)
openscience@royalsociety.org

Reviewer comments to Author:
Reviewer: 1

Comments to the Author(s)
File attached.

Reviewer: 2

Comments to the Author(s)

The paper reports on insights regarding uncertainty in and surrounding biophysical modeling. These insights were gained during an interdisciplinary research project on agricultural digitalization. I struggle with giving a recommendation on this manuscript. On the one hand, parts of the manuscript are insightful and contain useful distinctions and discussions. On the other hand, I fail to articulate what the main contribution is that the authors are trying to make. Moreover, I find large parts which dwell on interdisciplinarity tedious. If the authors can articulate more clearly in the introduction what the gap in the literature is that they are trying to address, and streamline the remainder of the manuscript around this contribution, I believe this will be a relevant paper worthy of publication. Let me elaborate this in more detail

First, I completely miss the point of section 2. How is section 2 adding to the contribution the paper is trying to make? In part, the issue might simply be that I lack a clear disciplinary identity

having been trained in and always having worked in interdisciplinary research. However, the issue I have with this section is also that it is not clearly connected to the focus of the paper: uncertainty in and surrounding biophysical modelling.

Second, on page 8, you correctly highlight that ambiguity is a kind of uncertainty next to epistemic and ontic (not ontological, but this is a fine philosophical point) uncertainty. This claim is not novel. Kwakkel, Walker, and Marchau (2010), with reference to Dewulf, Craps, Bouwen, Taillieu, and Pahl-Wostl (2005), explicitly include ambiguity within the nature dimension of Walker et al. (2003). However, since ambiguity is about knowing differently, it is an epistemic notion. Thus, the correct way is to make a distinction first between uncertainty as an attribute of the world (i.e. ontic or aleatoric uncertainty) and uncertainty due to our knowledge about the world (i.e. epistemic uncertainty). Within epistemic uncertainty, then, a further distinction can be made between not knowing enough and knowing different (i.e. ambiguity). As an aside, the nature dimension as conceptualized by Walker et al. originates in the first edition of Hacking (2006).

Third, the distinction between direct, indirect and contextual uncertainty is useful. However, I find it deeply confusing to call this different levels of uncertainty. Walker et al. (2003) already has a level dimension which refers to the degree or severity of uncertainty. To appropriate the level notion for something quite different, and closer to the idea of location in the Walker et al terminology, is confusing.

Fourth, the ideas developed in section 5, paragraph 3, are interesting. Walker et al have primarily used the level of uncertainty dimension as an organizing principle for the selection of ways of dealing with uncertainty. Correctly, the authors point out that the nature of uncertainty is also relevant. As pointed out by Dewulf repeatedly, if uncertainty is due to ambiguity more research will make controversies worse (Sarewitz, 2004). So, in case of ambiguity, collaborative sense making and joint fact finding are more appropriate.

Fifth, a recurring issue is that the paper interweaves two storylines. One on uncertainty in and surrounding biophysical modelling. The other containing primarily personal reflections on interdisciplinary research. The first is interesting and relevant. The second, I am unsure about its broader scientific relevance. Moreover, the interweaving of these two storylines makes the paper very dense in places and hard to follow.

Minor comment

p. 8 Line 40 "similar to brugnach et al", this statement is strange since Walker et al is older and if memory serves me well is being cited by Brugnach et al. So the logical order is that brugnach, following Wallker et al, is making a distinction between epistemic and ontic uncertainty.

Dewulf, A., Craps, M., Bouwen, R., Taillieu, T., & Pahl-Wostl, C. (2005). Integrated Management of Natural Resources: Dealing with Ambiguous Issues, Multiple Actors and Diverging Frames. *Water Science & Technology*, 52(6), 115-124.

Hacking, I. (2006). *The emergence of probability: a philosophical study of early ideas about probability, induction, and statistical inference* (2nd ed.). New York, NY: Cambridge University Press.

Kwakkel, J. H., Walker, W. E., & Marchau, V. A. W. J. (2010). Classifying and communicating uncertainties in model-based policy analysis. *International Journal of Technology, Policy and Management*, 10(4), 299-315. doi:10.1504/IJTPM.2010.036918

Sarewitz, D. (2004). How science makes environmental controversies worse. *Environmental Science & Policy*, 7, 385-403. doi:10.1016/j.envsci.2004.06.001

Walker, W. E., Harremoës, J., Rotmans, J. P., Van der Sluijs, J. P., Van Asselt, M. B. A., Janssen, P. H. M., & Krayer von Krauss, M. P. (2003). Defining Uncertainty: A Conceptual Basis for Uncertainty Management in Model-Based Decision Support. *Integrated Assessment*, 4(1), 5-17.

===PREPARING YOUR MANUSCRIPT===

===PREPARING YOUR REVISION IN SCHOLARONE===

Author's Response to Decision Letter for (RSOS-201511.R0)

See Appendix B.

Decision letter (RSOS-201511.R1)

Dear Dr Espig,

It is a pleasure to accept your manuscript entitled "Uncertainty in and around biophysical modelling: Insights from interdisciplinary research on agricultural digitalisation" in its current form for publication in Royal Society Open Science.

You can expect to receive a proof of your article in the near future. Please contact the editorial office (openscience_proofs@royalsociety.org) and the production office (openscience@royalsociety.org) to let us know if you are likely to be away from e-mail contact -- if

you are going to be away, please nominate a co-author (if available) to manage the proofing process, and ensure they are copied into your email to the journal.

on behalf of Dr Agnieszka Latawiec (Associate Editor) and Pete Smith (Subject Editor)
openscience@royalsociety.org

Appendix A

Review of RSOS-201511

This paper presents an extension of the framework presented by van der Bles, et al. (2019), addressing several issues regarding uncertainty modelling in interdisciplinary research. The authors begin by observing that "uncertainty" often is referred to by modelers and model users, but in a vague and/or ineffective fashion. They report insights from an interdisciplinary research program investigating uncertainties in measurements and models associated with technical developments in New Zealand agriculture. Overall, I think the paper presents several worthwhile ideas and, in the end, prescriptions for the conduct of interdisciplinary research. My comments are primarily oriented around some conceptual clarifications and questions about what conclusions can be drawn.

The authors do a creditable job of illustrating how different disciplinary and practice-domain "lenses" cast uncertainties in diverse ways, and their intent appears to be to provide a framework for integrating those diverse perspectives so that each of them carries legitimacy and weight. Such a framework is worth developing. However, my chief concern is that the paper does not directly address ways of dealing with what inevitably will be unshared orientations towards unknowns among members of an interdisciplinary team. I'll highlight two of these.

First, stakeholders may not share assessments of which unknowns are important or even relevant. To take a very simple example: Doctors are interested in the sensitivity and specificity of a diagnostic test (e.g., the probability that the test is positive, given that the patient has the disease). However, patients are more interested in its diagnosticity (e.g., the probability the patient has the disease, given a positive test). The first conditional probability can be quite high while the second one is low if the disease is rare. Or another example: Working as a statistical modeling consultant for a bank where my role was building models to predict bank losses, I was concerned about the accuracy of a model's predictions, but I found that the bank's quantitative modelers also were concerned about whether a model's predictions would be acceptable to the bank's senior managers. In this example the source of disagreement about relevance stems from differing views about the goals of the modeling exercise.

Second, the paper does not directly deal with what inevitably will be unshared unknowns among team members in many interdisciplinary research efforts. Complete consensus about what is unknown often is unattainable. For instance, if asked whether a health economist's model is sound an epidemiologist may shrug and say they don't know; and likewise if the health economist is asked whether the epidemiologist's disease transmission model is valid. Another kind of example: A model predicting economic performance may be regarded as seaworthy by one modeler but not by another because they disagree on whether the underlying economic processes are stationary or not. Interdisciplinary collaborations will require ways of dealing with unshared unknowns, such as the development of trust-based relationships among team members. It would be helpful if the authors could say something about this issue—It is one thing to regard a discipline as legitimate and respectable, but it is another to regard specific knowledge claims produced by a specialist in that discipline as trustworthy.

The "contextual uncertainty" concept is not entirely clear to me. It may need further explanation and examples. Taking a very simple example, an individual decision maker may believe they have strong evidence in support of an hypothesis, they may also regard the evidence as high-quality, and yet be unsure about what kind of decision-making procedure they should apply to that evidence (e.g., should they use a compensatory or non-compensatory combinatorial method?). Is this uncertainty about the decisional procedure an example of contextual uncertainty, or is there more to it than this?

One of the authors' first targets is the unequal standing of disciplines involved in interdisciplinary research, which they attribute to a combination of funding sources and the traditional distinctions between "hard" and "soft" sciences. However, they are not clear about what "equal standing" might consist of, or when it would be warranted. For instance, multi-authored works within a single discipline often do not contain equal contributions from all of the authors and in honest publications these inequalities are agreed upon by the authors and publicly announced in the publication. Is "equal standing" merely a question of regarding other disciplines as legitimate and respectable? Or is more required?

Appendix B

RSOS-201511: Response to reviewers

We thank both reviewers for their critical feedback and have responded to each of their comments below.

Reviewer 1

1. Dealing with different orientations towards unknowns

However, my chief concern is that the paper does not directly address ways of dealing with what inevitably will be unshared orientations towards unknowns among members of an interdisciplinary team.

Response: We appreciate the reviewer's comment, although the manuscript's scope is limited to uncertainties associated with biophysical modelling. Shared orientations towards unknowns may be a broader, or prior, question that is beyond our intended scope, depending on how one conceptualises unknowns, risks, indeterminacies, uncertainties etc. (see the narrowed scope in section 3.1, and footnotes 2 and 3). We chose to not further broaden the terminology here and focus on uncertainties.

We, nonetheless, responded as per below, and suggest that the *process* of building interdisciplinary collaboration and mutually shared understandings of uncertainty (e.g. by co-developing our proposed framework) is precisely the step required for moving beyond 'unshared orientations towards unknowns'. This does not necessarily mean developing entirely shared understandings, but at least mutual appreciation of different aspects of uncertainties/unknowns. Regarding interdisciplinary research as a process also means that 'inevitably unshared orientations' are not unchangeable and that teams can form shared understandings over time—which, in fact, is a key proposition in the manuscript. We included additional sections to highlight this aspect.

1.1. Stakeholders may not share assessments of which unknowns are important or even relevant.

Response: We added the following sentences to section 5: "Specifically, the outlined meta-categories of direct, indirect and contextual uncertainty allowed us to move towards a shared appreciation of the different facets of modelling uncertainties that our individual research projects addressed. Realising the complementarity between our approaches led to a better understanding of the aspects of biophysical modelling that might be uncertain, enabled us to systematically locate corresponding individual contributions, and created a mutually shared recognition of disciplinary strengths and limitations".

1.2. The paper does not directly deal with what inevitably will be unshared unknowns among team members in many interdisciplinary research efforts. ... Interdisciplinary collaborations will require ways of dealing with unshared unknowns, such as the development of trust-based relationships among team members. It would be helpful if the authors could say something about this issue.

Response: The reflections on the process of our interdisciplinary research in section 5 largely address this comment, but we added the following section for clarity: "(In this sense, the process of co-designing the three-level conceptual distinctions—perhaps more

so than the framework itself—contributed to the creation of discursive and interactive spaces to explore knowledge production beyond disciplinary boundaries,) which involved building a trusted team culture that allowed for constructive dialogue.”

2. Contextual Uncertainty

The "contextual uncertainty" concept is not entirely clear to me. It may need further explanation and examples.

Response: Further to the definition, Box 1, and reference to the similar notion of ‘situational uncertainty’, we have added an example for agricultural biophysical modelling to section 4: “In the context of biophysical modelling in the agricultural sector, contextual uncertainty may manifest for some actors if it is unclear how modelled information will be used in environmental regulation, regardless of direct and indirect uncertainties associated with the corresponding model (e.g. PCE 2018).”

3. Equal standing

One of the authors' first targets is the unequal standing of disciplines involved in interdisciplinary research, which they attribute to a combination of funding sources and the traditional distinctions between "hard" and "soft" sciences. However, they are not clear about what "equal standing" might consist of, or when it would be warranted. For instance, multi-authored works within a single discipline often do not contain equal contributions from all of the authors and in honest publications these inequalities are agreed upon by the authors and publicly announced in the publication. Is "equal standing" merely a question of regarding other disciplines as legitimate and respectable? Or is more required?

Response: Our argument goes beyond the authorship of individual publications. We are here concerned with the *process* of interdisciplinary research and the equally valued contribution of disciplines throughout, including, as we argue in sections 2 and 5, early collaborative problem framing. In sections 5, the last paragraph also refers back to the notion of ‘interdisciplinary logic of ontology’, which includes “not merely engaging in research that is framed as interdisciplinary research, but to foster the ‘cultivation of interdisciplinary subjectivities and skills’”. These are crucial aspects of an equal standing between disciplines.

We further define ‘equal standing’ in section 2 as follows—incl. a sentence added in response to Reviewer 2 (see point 1): ‘This requires cultivating an appreciation for different disciplinary insights that are of equal epistemic standing, rather than regarding some disciplines as merely ‘supporting’ core research. ... Brondizio (2017) therefore, first and foremost, regards interdisciplinarity as collaborative problem framing and a process of critical thinking and reflection. ... Programmes that seek to holistically approach modelling uncertainties are one such interdisciplinary research setting, and we argue that equally valued disciplinary contributions must be included already during problem framing and design stages, which should then translate into adequate resourcing and integration of different research areas.’”

We further added a corresponding sentence to section 5, last paragraph: “This should include re-evaluating unequal disciplinary standings that can, for example, lead to fewer funding opportunities for research on the sociocultural facets of modelling uncertainty and agricultural digitalisation as compared to their biophysical aspects.”

Reviewer 2

1. Connection between two lines of argumentation

On the one hand, parts of the manuscript are insightful and contain useful distinctions and discussions. On the other hand, I fail to articulate what the main contribution is that the authors are trying to make. Moreover, I find large parts which dwell on interdisciplinarity tedious. If the authors can articulate more clearly in the introduction what the gap in the literature is that they are trying to address, and streamline the remainder of the manuscript around this contribution, I believe this will be a relevant paper worthy of publication.

First, I completely miss the point of section 2.

Fifth, a recurring issue is that the paper interweaves two storylines. One on uncertainty in and surrounding biophysical modelling. The other containing primarily personal reflections on interdisciplinary research. The first is interesting and relevant. The second, I am unsure about its broader scientific relevance. Moreover, the interweaving of these two storylines makes the paper very dense in places and hard to follow.

Response: We appreciate that some readers’ interest might gravitate towards either of the two interwoven lines of argumentation (cf. Reviewer 1’s primary concern with interdisciplinarity). Nonetheless, we regard both as complimentary since understanding modelling uncertainty holistically requires inter-/transdisciplinary approaches that come with their own challenges. The reviewer appears to affirm the connection between these insights by noting that the ‘ideas developed in section 5, paragraph 3, are interesting’ (see point 4.).

We further clarified the connection between the lines of argumentation, including the paper’s main contribution and the relevance of section 2, by adding the following sections:

- section 1, paragraph 3: “The article’s main contribution is therefore to demonstrate that holistically understanding modelling uncertainties requires meaningful collaboration between researchers from a range of disciplines. However, establishing successful interdisciplinary approaches is usually a laborious process that involves overcoming numerous practical and conceptual challenges. We present these insights to prompt discussion among researchers and practitioners about how modelling uncertainties may be addressed in a systematic fashion and to reflect on the underlying processes of interdisciplinary research, which are often overlooked.”
- section 1, last paragraph: “(The next section reflects on interdisciplinary agricultural research and the challenges of equal disciplinary standing) to contextualise our insights.”
- section 2, paragraph 1: “It is thus important to consider the contexts in which interdisciplinary research of modelling uncertainties takes place in order to assess, and improve, projects’ efficacy.”

- section 2, paragraph 4: “We regard this type of deeper engagement with interdisciplinary research as an important component of programmes that seek to address the various facets of modelling uncertainties.”
- section 2, last paragraph: “Programmes that seek to holistically approach modelling uncertainties are one such interdisciplinary research setting, and we argue that equally valued disciplinary contributions must be included already during problem framing and design stages, which should then translate into adequate resourcing and integration of different research areas.”

Re: the suspected scientific irrelevance of insights related to interdisciplinary research, it is worth noting that this is a field of scientific research and practice in its own right. For instance, we refer to Gabrielle Bammer and colleagues’ work on Integration and Implementation Sciences. Integrated and interdisciplinary approaches are also increasingly employed across a range of research programmes; we noted our own organisation’s digital agriculture programme as one example.

Re: the manuscript’s density, we reviewed the entire manuscript for readability and made minor editorial changes, including:

- deleted some examples (e.g. Anthropocene concept in section 5) and shortened several sentences
- deleted original footnotes 2, 8 and 10.

2. Ambiguity

You correctly highlight that ambiguity is a kind of uncertainty next to epistemic and ontic (not ontological, but this is a fine philosophical point) uncertainty. This claim is not novel. Kwakkel, Walker, and Marchau (2010), with reference to Dewulf, Craps, Bouwen, Taillieu, and Pahl-Wostl (2005), explicitly include ambiguity within the nature dimension of Walker et al. (2003). However, since ambiguity is about knowing differently, it is an epistemic notion. Thus, the correct way is to make a distinction first between uncertainty as an attribute of the world (i.e. ontic or aleatoric uncertainty) and uncertainty due to our knowledge about the world (i.e. epistemic uncertainty). Within epistemic uncertainty, then, a further distinction can be made between not knowing enough and knowing different (i.e. ambiguity).

Response: In outlining a social science perspective and relevant literature in section 3.1, we do not propose that referring to ambiguity as a third kind of uncertainty is a novel claim. The paragraph in question centres around Brugnach et al. (2008)* who explicitly refer to ‘Ambiguity: Uncertainty of a Third Kind’ and note: “we consider ambiguity as a third kind or nature of uncertainty, along with ontological and epistemic uncertainty, rather than just a source of uncertainty. For us, the relevant dimension for ambiguity is not the one from complete knowledge to complete ignorance, but something ranging from unanimous clarity to total confusion caused by too many people voicing different but still valid interpretations (Dewulf et al. 2005).”

We did not change the relevant paragraph since (i) it is, in our reading, an accurate summary of Brugnach et al. (2008), and (ii) in line with the social dimensions and relational perspective we outline in regards to indirect uncertainty in section 5. We thus

also retain the notion of ontological uncertainty, which is similarly adopted by several other authors, but do appreciate the philosophical nuances between ontic and ontological.

*Please note that Dewulf, Pahl-Wostl and Taillieu are three of the four authors in Brugnach et al. (2008), so we assume that the authors are well aware of Dewulf, Craps, Bouwen, Taillieu, and Pahl-Wostl (2005) as used by Kwakkel et al. (2010).

3. Levels of uncertainty

The distinction between direct, indirect and contextual uncertainty is useful. However, I find it deeply confusing to call this different levels of uncertainty. Walker et al. (2003) already has a level dimension which refers to the degree or severity of uncertainty. To appropriate the level notion for something quite different, and closer to the idea of location in the Walker et al terminology, is confusing.

Response: We appreciate that reference to ‘levels’ of uncertainty may create some confusion with the three dimensions Walker et al. (2003) propose: location, level and nature of uncertainty—perhaps a case in point of the semantic ambiguity (i.e. linguistic uncertainty) we note in section 3. However, our manuscript is an explicit response to the *RSOS* publication by van der Bles et al., who refer to ‘levels of uncertainty’ to describe direct and indirect uncertainty (see section 3.3. in van der Bles et al. 2019).

We have, therefore, maintained the terminology of levels of uncertainty, but added the following footnote 7 to section 4 for clarification: “Van der Bles et al.’s (2019) conceptualisation of ‘levels’ of uncertainty is somewhat different to Walker et al. (2003), who propose it as one of three dimensions to differentiate uncertainties (see section 3.2). We here follow van der Bles and co-authors when referring to levels of uncertainty.”

We further deleted references to ‘locations’ of uncertainty throughout the manuscript to avoid confusion (excluding the review of Walker et al. 2003) and refer to ‘levels’ or simply ‘different’ uncertainties.

4. Comment on process and ambiguity

Fourth, the ideas developed in section 5, paragraph 3, are interesting. Walker et al have primarily used the level of uncertainty dimension as an organizing principle for the selection of ways of dealing with uncertainty. Correctly, the authors point out that the nature of uncertainty is also relevant. As pointed out by Dewulf repeatedly, if uncertainty is due to ambiguity more research will make controversies worse (Sarewitz, 2004). So, in case of ambiguity, collaborative sense making and joint fact finding are more appropriate.

Response: We understand this comment to affirm the argumentation in Section 5 that refers to ‘collaborative problem framing’, which we develop partly in response to the fact that “ambiguity can emerge from multiple knowledge frames through which uncertainty gains meaning” (see section 3.1). We regard the process of collaborative problem framing as involving collaborative sense making and the joint generation of knowledge. In response to the reviewer’s comment, we added Brugnach et al. (2011) to section 3.1.

While the notion that ‘more research makes controversies worse’ warrants further investigation in relation to biophysical modelling, we cannot address this point here beyond the brief note in section 3.1. Both primary authors have investigated the role of scientific research in environmental controversies and health-related questions. This aspect of uncertainty was thus considered but consciously not addressed further. Suffice to note that, as we propose, inter- or transdisciplinary programmes that include a range of natural, social and applied scientific approaches are required to address the various forms of uncertainty associated with biophysical modelling—and social controversies more broadly.

5. Minor comment

p. 8 Line 40 “similar to brugnach et al”, this statement is strange since Walker et al is older and if memory serves me well is being cited by Brugnach et al.

Response: The use of ‘Similar to Brugnach et al. (2008), Walker et al. (2003)’ was not intended to imply a (false) chronological order but to establish a connection between sections 3.1 and 3.2. We deleted ‘Similar to Brugnach et al. (2008)’ to avoid any confusion.

Additional revisions

- we added Schikowitz (2020) as a reference in several places.